# Perioperative Red Cell Line Trend following Robot-Assisted Radical Prostatectomy for Prostate Cancer

**DOI:** 10.3390/medicina58111520

**Published:** 2022-10-25

**Authors:** Francesco Di Bello, Ernesto Di Mauro, Claudia Collà Ruvolo, Massimiliano Creta, Roberto La Rocca, Giuseppe Celentano, Marco Capece, Luigi Napolitano, Simone Morra, Gabriele Pezone, Francesco Passaro, Ciro De Luca, Francesco Mangiapia, Nicola Logrieco, Pasquale Buonanno, Giuseppe Servillo, Ciro Imbimbo, Vincenzo Mirone, Nicola Longo, Gianluigi Califano

**Affiliations:** Department of Neuroscience, Reproductive and Odontostomatological Sciences, University of Naples “Federico II”, Via Pansini, 80138 Naples, Italy

**Keywords:** PCa, RARP, blood loss, hemoglobin, red blood cell, hematocrit, surgery, surgical oncology

## Abstract

*Background and Objective*: Blood loss represents a long-standing concern of radical prostatectomy (RP). This study aimed to assess how red line cell values changed following robot-assisted radical prostatectomy (RARP) for prostate cancer (PCa). *Materials and Methods*: The blood panels of 453 consecutive PCa patients undergoing RARP at a single tertiary academic referral center, from September 2020 to April 2022, were reviewed. Data from 363 patients with the blood panel available for the following timeframe: within seven days before surgery, six hours after surgery, and the first three postoperative days, were analyzed. Specifically, hemoglobin (Hb, g/dL), red blood cells (RBCs, ×10^6^/μL), and hematocrit (HCT, %) trends were collected. *Results*: Considering the Hb trend, the median values in the preoperative day, postoperative day (POD) 2, and POD 3 are 14.7 (interquartile range (IQR) = 13.9–15.4), 12.1 (IQR = 11.2–12.9), and 12.2 (IQR = 11.2–13.1), respectively. The ∆ between preoperative day and POD 2 is 2.5 (IQR = 1.8–3.2) (*p* < 0.001). Considering the RBCs trend, the median values in the preoperative day, POD 2, and POD 3 are 4.9 (IQR = 4.7–5.3), 4.1 (IQR = 3.8–4.4), and 4.1 (IQR = 3.8–4.5), respectively. The ∆ between preoperative day and POD 2 is 0.9 (IQR = 0.6–1.1) (*p* < 0.001). Considering the HCT trend, the median values in the preoperative day, POD 2, and POD 3 are 44.4 (IQR = 41.7–46.6), 36.4 (IQR = 33.8–38.9), and 36.1 (IQR = 33.5–38.7), respectively. The ∆ between preoperative day and POD 2 is 7.8 (IQR = 5.2–10.5) (*p* < 0.001). *Conclusions*: Overall, patients undergoing RARP experience a significant, but clinically limited, decline in red line cell values between the preoperative time and the second day post-surgery. These observations are important to provide physicians with knowledge of the expected postoperative course and, thus, to improve the quality of patient care.

## 1. Introduction

Radical prostatectomy (RP) is the standard surgical treatment for patients with clinically localized prostate cancer (PCa) [1,2]. The procedure involves removing the entire prostate with its capsule intact and seminal vesicles, followed by vesicourethral anastomosis [1]. RP can be performed by open (ORP) or minimally invasive approaches [3].

Laparoscopic (LRP) and robot-assisted RP (RARP) are progressively emerging as the preferred options [1,4]. The use of robots combines the improving ergonomics and surgical margin control, lowering the perioperative morbidity rate compared to conventional laparoscopic techniques [3,5]. Moreover, RARP shows a significant impact on the quality of patient care management, reducing intraoperative blood loss, length of stay, and postoperative pain, speeding up the patient’s recovery [6,7,8].

Historically, blood loss represented a long-standing concern of RP [6,7]. Several studies assessed the benefits of RARP on blood loss control in PCa patients during the intraoperative and postoperative period [6,7,9]. To the best of our knowledge, no previous study primarily investigated the trend of red line cells in the postoperative days (POD) following RARP. Patients’ management, length of stay, and overall healthcare costs of surgery are strictly related to the impairment of perioperative red line cells trend (10).

The aim of this study was to assess how red line cell values changed following RARP for PCa.

## 2. Materials and Methods

### 2.1. Study Population

The blood panels of PCa patients undergoing RARP at a single tertiary academic referral center, from September 2020 to April 2022, were reviewed. The procedures were performed by three independent surgeons with similar high-volume experience in minimally invasive surgery. The RARP was performed with or without lymph node dissection according to the European Association of Urology guidelines indication (11). The institutional database was searched for PCa patients undergoing RARP with the blood panel available for the following timeframe: within seven days before surgery, six hours after surgery, and POD1, POD2, and POD3 mornings. Specifically, hemoglobin (Hb, g/dL), red blood cells (RBCs, ×10^6^/μL), and hematocrit (HCT, %) trends were assessed. RARP was performed using a standard 26° Trendelenburg position. The same liquid management protocol was applied: a 4 mL/proKg/h infusion of crystalloids until the patient awakened, followed by an 8–10 mL/proKg/h until POD 1.

### 2.2. Statistical Analysis

Descriptive statistics included means with the standard deviation (SD) and medians with the interquartile ranges (IQR) for continuously coded variables. The Wilcoxon sign rank test for paired samples was used to compare continuous nonparametric variables. In all statistical analyses, the R software environment for statistical computing and graphics (R version 3.6.1) (R Development Core Team, Auckland, New Zealand) was used. All tests were two-sided, with a level of significance set at *p* < 0.05.

## 3. Results

A total of 453 blood panel patients were examined. Of all, according to the inclusion criteria, 363 (80.1%) patients’ data were included in the final analyses (Table 1).

Considering the Hb trend (Figure 1A), the median values in the preoperative day, POD1, POD 2, and POD 3 are 14.7 (IQR = 13.9–15.4), 12.6 (IQR = 11.8–13.4), 12.1 (IQR = 11.2–12.9). and 12.2 (IQR = 11.2–13.1), respectively. The ∆ between preoperative day and POD 1 is 2.0 (*p* < 0.001). The ∆ between preoperative day and POD 2 is 2.5 (*p* < 0.001). Conversely, ∆ between POD 2 and POD 3 is 0.5 (*p* = 0.6).

Considering the RBCs trend (Figure 1B), the median values in the preoperative day, POD1, POD 2, and POD 3 are 4.9 (IQR = 4.7–5.3), 4.3 (IQR = 4.0–4.6), 4.1 (IQR = 3.8–4.4), and 4.1 (IQR = 3.8–4.5), respectively. The ∆ between preoperative day and POD 1 is 0.6 (*p* < 0.001). The ∆ between preoperative day and POD 2 is 0.9 (*p* < 0.001). Conversely, ∆ between POD 2 and POD 3 is 0.2 (*p* = 0.3).

Considering the HCT trend (Figure 1C), the median values in the preoperative day, POD1, POD 2, and POD 3 are 44.4 (IQR = 41.7–46.6), 38 (IQR= 35.5–40.1), 36.4 (IQR = 33.8–38.9), and 36.1 (IQR = 33.5–38.7), respectively. The ∆ between preoperative day and POD 1 is 6.3 (*p* < 0.001). The ∆ between preoperative day and POD 2 is 7.8 (*p* < 0.001). Conversely, ∆ between POD 2 and POD 3 is 1.5 (*p* = 0.5).

## 4. Discussion

Historically, blood loss represented a long-standing concern of RP [7,9]. To the best of our knowledge, no previous study primarily investigated the trend of red line cells in the postoperative days following RARP. The aim of the current study was to assess how red line cell values changed following RARP for PCa.

From our analysis, it emerges that three days after surgery, the median decrease in Hb, RBCs, and HCT values is 3 g/dL, 1.1 × 10^6^/μL, and 9.3%, respectively. The median drop in all the parameters analyzed is significant up to POD2. However, the trend stabilizes on POD3. These results highlight important data that fill a knowledge gap in the literature and support the growing centers.

A thorough evaluation of the presented data results in the following clinical observations. First, a limited three point hemoglobin loss following RARP for PCa should not cause concern for the risk of active bleeding. Conservative management is, therefore, recommended, according to the patient’s comorbidities. Second, the downward trend lasts until the second or third POD, and then quickly stabilizes. Longer decreases over time, on the other hand, should be assessed with imaging techniques. However, the above observations should be corroborated after the consideration of other important clinical and intraoperative information, such as anticoagulant and/or antiaggregant drugs assumption, lymph nodes dissection, or intraoperative blood loss. Moreover, a prospective evaluation should also be performed in order to strengthen the robustness of these considerations.

PCa patient’s care goes far beyond disease diagnosis and treatment [10]. The correct perioperative management of patients undergoing RARP is a crucial aspect of the care pathway. The implications are significant for patients in the first place and more generally for the optimization of health resources. The lack of knowledge on the perioperative management aspects can lead to unjustified concerns and inadequate strategic choices.

Indeed, early discharge of patients after surgery should be recommended in order to minimize healthcare costs, without compromising patient safety [11,12]. For example, Abaza et al. demonstrated the clinical safety of offering same-day discharge (SDD) protocol to 500 RARP patients associated with cost savings and an open hospital bed for patients with more acute conditions [13]. Similarly, Ploussard et al. observed the safety and the cost reduction when the SDD was offered to PCa patients treated with RARP in the context of enhanced recovery after surgery protocols [14]. According to our results, and the results of Abaza and Plussard et al., an early discharge of patients without a straight blood parameter evaluation might be recommended without compromising patients’ safety. However, the presence of specific patients and surgical-relate conditions that might increase the risk of clinically significant bleeding must be never underestimated. To date, in our daily clinical practice, we prefer to keep patients under our observations to facilitate the follow-up of blood tests to ensure we dismiss patients in the safest conditions. We plan to change the patient management protocol in the future, after findings based on prospective and multicenter results.

Taken together, this study focused on the overall red line cells pattern in the postoperative period following RARP for PCa. The resulting observations are representative of the general population. However, several factors can independently influence changes in red line cells after surgery, generically related to the patient, surgery, intra and perioperative anesthetic, and infusion management. Future studies with prospective design and multicenter involvement are needed to evaluate the significance of these variables on postoperative red line cells changes in patients undergoing RARP for PCa.

Despite novel and important observations, several limitations may be applicable to our study. First, other clinical (such as tumor stage and grade, antiaggregant or anticoagulant drugs assumption, or body mass index) and intraoperative data (such as blood loss, intraoperative time, or lymph node dissection data) were not included in the current analyses. However, the aim of the current study was to provide a preliminary analyses of blood parameter trends in RARP patients, regardless of other confounders. Second, the retrospective and single-center nature of the study represents in itself a limitation. Future prospective and multicenter studies should be conducted in order to corroborate or reject our preliminary findings. Despite the above limitations, we were the first to investigate blood parameter trends in immediate perioperative time of PCa patients that underwent RARP.

## 5. Conclusions

Overall, patients undergoing RARP experience a significant, but clinically limited, decline in red line cells values in the postoperative period. Specifically, a reduction by roughly three points of Hb is recorded between preoperative time and the second day after surgery. However, the negative trend seems to quickly stabilize after the third day post-surgery. These observations are important to provide physicians with knowledge of the expected postoperative course and, thus, to improve the quality of patient care.

## Figures and Tables

**Figure 1 medicina-58-01520-f001:**
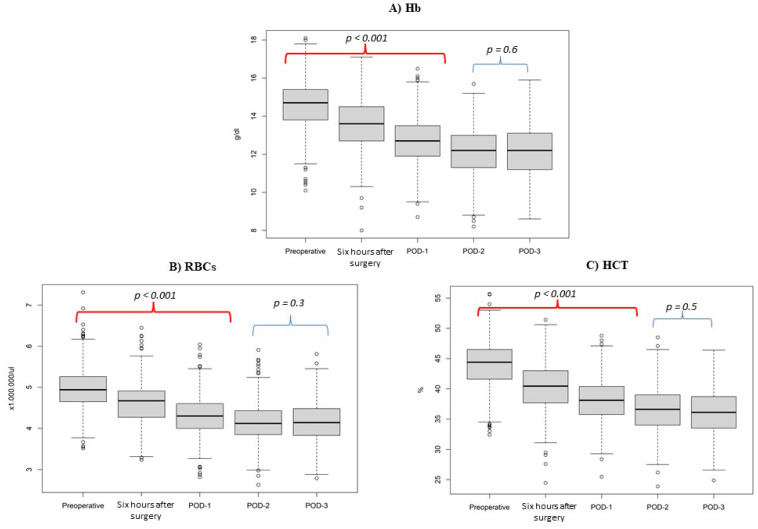
(**A**): Hemoglobin (Hb), (**B**): red blood cell (RBCs), and (**C**): hematocrit (HCT) trends in 363 prostate cancer patients undergoing robot-assisted radical prostatectomy, from September 2020 to April 2022, at preoperative day, six hours after surgery, and postoperative day (POD) 1, POD 2, and POD 3. Boxes denote the interquartile range. The solid black horizontal bar denotes the median within each perioperative time. Whiskers denote the 95% range of the distribution of red line cell values. The open circles denote outlier values.

**Table 1 medicina-58-01520-t001:** Hemoglobin (Hb), red blood cell (RBCs), and hematocrit (HCT) in 363 prostate cancer patients undergoing robot-assisted radical prostatectomy, from September 2020 to April 2022, at preoperative day, postoperative day (POD)1, POD 2, and POD 3.

		Preoperative Day	POD 1	POD 2	POD 3	∆ Preoperative– POD 1	*p* Value	∆ Preoperative– POD 2	*p* Value	∆ POD 2– POD 3	*p* Value
Hb (g/dL)	Mean (SD)	14.6 (0.071)	12.6 (0.064)	12.1 (0.068)	12.1 (0.071)	2.0	<0.001	2.5	<0.001	0.6	0.6
	Median (IQR)	14.7 (13.9–15.4)	12.6 (11.8–13.4)	12.1 (11.2–12.9)	12.2 (11.2–13.1)	2.1		2.5		0.5	
RBCs (×10^6^/μL)	Mean (SD)	5 (0.028)	4.3 (0.025)	4.1 (0.026)	4.1 (0.027)	0.7	<0.001	0.9	<0.001	0.2	0.3
	Median (IQR)	4.9 (4.7–5.3)	4.3 (4.0–4.6)	4.1 (3.8–4.4)	4.1 (3.8–4.5)	0.6		0.9		0.2	
HCT (%)	Mean (SD)	44.2 (0.213)	37.9 (0.186)	36.3 (0.209)	36.1 (0.211)	6.3	<0.001	7.9	<0.001	2.0	0.5
	Median (IQR)	44.4 (41.7–46.6)	38 (35.5–40.1)	36.4 (33.8–38.9)	36.1 (33.5–38.7)	6.3		7.8		1.5	

Abbreviations: IQR = interquartile range; SD = standard deviation.

## Data Availability

Data available on request due to restrictions eg privacy or ethical.

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
