# Peer review of "Perioperative Red Cell Line Trend following Robot-Assisted Radical Prostatectomy for Prostate Cancer"

_medicina, 2022, doi:10.3390/medicina58111520_

Round 1
Reviewer 1 Report (Previous Reviewer 1)
The authors addressed my points raised
Author Response
We thank the Reviewer for the comment.
Reviewer 2 Report (New Reviewer)
I congratulate the authors for the manuscript.
It is interesting and well-written.
There are some concern the authors should address.
It is RARP without pelvic lymphadenectomy?
There are some English grammar mistakes that should be corrected.
-Abstract
Add IQR meaning
Please change the location/ “ RARP is changing the way we think about surgical management of PCa patients” in abstract and full text. Conclusion should be related to the objective and results.
Typo a “.” After period is missing
-Introduction
The sentence “Patients’ management, length of stay and overall healthcare costs of surgery are strictly related to the impairment of perioperative red line cells trend.” Must be referenced.
-Results
Figure, for easier understanding of the figure the box-plot should be described i.e. the boxes represent the median and IQR and 95% percentiles.
-M and M
Why did you use the Kruskall Wallis test and not the rank sum test?
The author found no difference between POD 2 and 3. Was there significant difference between preoperative and day 1, and between POD 2 and POD 3?
-Discussion
“First, a limited 3-point hemoglobin loss following RARP for PCa should not cause concern about the risk of active bleeding”. However, in the study there was no clinical or complication information other than blood values to state so.
“ the downward trend lasts until the second-third POD and then quickly stabilizes”. Longer decreases over time, on the other hand, should be assessed with imaging techniques. However, 3 days is the limit time point were the study measured and correlation with clinical evolution was not performed. This point should be clarified.
Is it interesting that other studies performed same day discharge. What is the authors proposal based in this manuscript information?
Author Response
is RARP without pelvic lymphadenectomy?
Answer: We thank the Reviewer for the pertinent comment. The lymph nodes dissection was performed according to the EAU guidelines indications. However, in the current preliminary study, we did not include the clinical and intraoperative data in order to describe the perioperative trend of blood test after RARP, regardless of any confounder. Thus, we better specified this point in the new version of the manuscript which now reads as follow:
“The RARP was performed with or without lymph nodes dissection according to the EAU guidelines indication”
“Despite novel and important observations, several limitations may be applicable to our study. First, other clinical (such as tumor stage and grade, antiaggregant or anticoagulant drugs assumption or body mass index) and intraoperative data (such us blood loos, intraoperative time or lymph node dissection data) were not included in the current analyses.”
There are some English grammar mistakes that should be corrected.
Answer: We thank the Reviewer for the comment. We reviewed the manuscript with a mother tongue colleague and corrected the grammar mistakes.
-Abstract
Add IQR meaning
Please change the location/ “ RARP is changing the way we think about surgical management of PCa patients” in abstract and full text. Conclusion should be related to the objective and results.
Typo a “.” After period is missing
Answer: We thank the Reviewer for the pertinent comments.
- Regarding the IQR: we specified the abbreviation in the abstract section, which now reads as follow:
“Considering the Hb trend the median values in the preoperative day, postoperative day (POD) 2 and POD 3 were 14.7 (Interquartile range [IQR]=13.9–15.4),..”
- Regarding the sentence “RARP is changing the way we think about surgical management of PCa patients” and conclusion:
We excluded the quoted sentence from the abstract section and from the main text, Moreover, we modified the conclusion section in both the abstract and in the full text in the following way:
Abstract: “Overall, patients undergoing RARP experienced a significant but clinically limited decline in red line cell values between the preoperative time and the second-day post-surgery. These observations are important to provide physicians with knowledge of the expected postoperative course and thus to improve the quality of patient care.”
Full text: “Overall, patients undergoing RARP experienced a significant, but clinically limited decline in red line cells values in the postoperative period. Specifically, a reduction of roughly three points of Hb was recorded between preoperative time and the second-day after surgery. However, the negative trend seems to quickly stabilizes after the third day post-surgery. These observations are important to provide physicians with knowledge of the expected postoperative course and thus to improve the quality of patient care.”
- Regarding the typo mistake: we went over the entire text and added the “.” after the period when missing.
-Introduction
The sentence “Patients’ management, length of stay and overall healthcare costs of surgery are strictly related to the impairment of perioperative red line cells trend.” Must be referenced.
- Answer: We thank the Reviewer for the pertinent comments. We added the reference (PMID: 34131882), where Porcaro et al. revealed that perioperative blood loss associated with blood transfusion resulted in a delayed discharge and a increased hospital costs.
-Results
Figure, for easier understanding of the figure the box-plot should be described i.e. the boxes represent the median and IQR and 95% percentiles.
Answer: We thank the Reviewer for the pertinent comments. We better described the boxplots in the figure legend, which now reads as follows:
“Figure 1. Hemoglobin (Hb), Red blood cell (RBCs) and Hematocrits (HCT) trends in 363 prostate cancer patients undergoing robot-assisted radical prostatectomy, from September 2020 to April 2022, at preoperative day, six hours after surgery, post operative day (POD) 1, POD 2 and POD 3. Boxes denote the interquartile range. The solid black horizontal bar denotes the median within each perioperative time. Whiskers denote the 95% range of the distribution of red line cell values. The open circles denote outlier values.”
-M and M
Why did you use the Kruskall Wallis test and not the rank sum test?
The author found no difference between POD 2 and 3. Was there significant difference between preoperative and day 1, and between POD 2 and POD 3?
Answer: We thank the Reviewer for the pertinent comments.
- Regarding the statistical test used: we used the Wilcoxon sign rank test for paired samples. We corrected the new version of the manuscript as follows:
“Descriptive statistics included means with the standard deviation (SD) and medians with the interquartile ranges (IQR) for continuously coded variables. The Wilcoxon sign rank test for paired samples was used to compare continuous nonparametric variables.”
- Regarding the differences between preoperative time and POD 1 and between POD2 and POD3: We included the value recorded at POD1 and the relative Delta between preoperative time and POD1 in the results and in Table 1. Regarding the DELTA between POD 2 and POD 3, the relative data are already reported in the results and table in the previous version of the manuscript.
“Considering the Hb trend (Figure 1A), the median values in the preoperative day, POD1, POD 2 and POD 3 were 14.7 (IQR = 13.9–15.4), 12.6 (IQR = 11.8 – 13.4), 12.1 (IQR = 11.2–12.9) and 12.2 (IQR = 11.2–13.1), respectively. The ∆ between preoperative day and POD 1 was 2.0 (p < 0.001). The ∆ between preoperative day and POD 2 was 2.5 (p < 0.001). Conversely, ∆ between POD 2 and POD 3 was 0.5 (p = 0.6).
Considering the RBCs trend (Figure 1B), the median values in the preoperative day, POD1, POD 2 and POD 3 were 4.9 (IQR= 4.7–5.3), 4.3 (IQR = 4.0 - 4.6), 4.1 (IQR = 3.8–4.4) and 4.1 (IQR = 3.8–4.5), respectively. The ∆ between preoperative day and POD 1 was 0.6 (p < 0.001). The ∆ between preoperative day and POD 2 was 0.9 (p < 0.001). Conversely, ∆ between POD 2 and POD 3 was 0.2 (p = 0.3).
Considering the HCT trend (Figure 1C), the median values in the preoperative day, POD1, POD 2 and POD 3 were 44.4 (IQR = 41.7–46.6), 38 (IQR= 35.5 - 40.1), 36.4 (IQR = 33.8–38.9) and 36.1 (IQR = 33.5–38.7), respectively. The ∆ between preoperative day and POD 1 was 6.3 (p < 0.001). The ∆ between preoperative day and POD 2 was 7.8 (p < 0.001). Conversely, ∆ between POD 2 and POD 3 was 1.5 (p = 0.5).”
|
|
Preoperative day |
POD 1 |
POD 2 |
POD 3 |
∆ Preoperative – POD 1 |
p value |
∆ Preoperative – POD 2 |
p value |
∆ POD 2 – POD 3 |
p value |
|
|
Hb (g/dl) |
Mean (SD) |
14.6 (0.071) |
12.6 (0.064) |
12.1 (0.068) |
12.1 (0.071) |
2.0 |
<0.001 |
2.5 |
<0.001 |
0.6 |
0.6 |
|
Median (IQR) |
14.7 (13.9–15.4) |
12.6 (11.8-13.4) |
12.1 (11.2–12.9) |
12.2 (11.2–13.1) |
2.1 |
2.5 |
0.5 |
||||
|
RBCs (x106/uL)
|
Mean (SD) |
5 (0.028) |
4.3 (0.025) |
4.1 (0.026) |
4.1 (0.027) |
0.7 |
<0.001 |
0.9 |
<0.001 |
0.2 |
0.3 |
|
Median (IQR) |
4.9 (4.7–5.3) |
4.3 (4.0-4.6) |
4.1 (3.8–4.4) |
4.1 (3.8–4.5) |
0.6 |
0.9 |
0.2 |
||||
|
HCT (%) |
Mean (SD) |
44.2 (0.213) |
37.9 (0.186) |
36.3 (0.209) |
36.1 (0.211) |
6.3 |
<0.001 |
7.9 |
<0.001 |
2.0 |
0.5 |
|
Median (IQR) |
44.4 (41.7–46.6) |
38 (35.5-40.1) |
36.4 (33.8–38.9) |
36.1 (33.5–38.7) |
6.3 |
7.8 |
1.5 |
||||
-Discussion
“First, a limited 3-point hemoglobin loss following RARP for PCa should not cause concern about the risk of active bleeding”. However, in the study there was no clinical or complication information other than blood values to state so.
“the downward trend lasts until the second-third POD and then quickly stabilizes”. Longer decreases over time, on the other hand, should be assessed with imaging techniques. However, 3 days is the limit time point were the study measured and correlation with clinical evolution was not performed. This point should be clarified.
Is it interesting that other studies performed same day discharge. What is the authors proposal based in this manuscript information?
Answer: We thank the Reviewer for the pertinent comments.
- Regarding the sentences “First, a limited 3-point hemoglobin loss following RARP for PCa should not cause concern about the risk of active bleeding” and “the downward trend lasts until the second-third POD and then quickly stabilizes”: We agree with the Reviewer’s comment. Thus, we implemented the discussion, which now reads as follows:
- “First, a limited 3-point hemoglobin loss following RARP for PCa should not cause concern about the risk of active bleeding. Conservative management is therefore recommended, according to the patients’ comorbidities. Second, the downward trend lasts until the second-third POD and then quickly stabilizes. Longer decreases over time, on the other hand, should be assessed with imaging techniques. However, the above observations should be corroborated after the consideration of other important clinical and intraoperative information, such as anticoagulant and/or antiaggregant drugs assumption, lymph nodes dissection or intraoperative blood loss. Moreover, a prospective evaluation should also be performed in order to strengthen the robustness of these considerations.”
- “First, other clinical (such as tumor stage and grade, antiaggregant or anticoagulant drugs assumption or body mass index) and intraoperative data (such us blood loos, intraoperative time or lymph node dissection data) were not included in the current analyses.”
- Regarding the same day discharge protocol, we deeply discussed this point in the new version of the manuscript, which now reads as follows: “Indeed, early discharge of patients after surgery should be recommended in order to minimize healthcare costs, without compromising patient safety [10,11]. For example, Abaza et al. demonstrated the clinical safety in offering same day discharge (SDD) protocol to 500 RARP patients associated with a cost savings and an open hospital bed for patients with more acute conditions [10,12]. Similarly, Ploussard et al. observed the safety and the cost reduction when the SDD was offered to PCa patients treated with RARP in the context of enhanced recovery after surgery protocols [11]. According to our results and to Abaza and Plussard et al results, an early discharge of patients without a straight blood parameter evaluation might be recommended without compromise patients safety. However, it must be never underestimated the presence of specific patients and surgical related conditions that might increase the risk of clinically significant bleeding.”
We hope that the Reviewer will find this reply satisfactory.

Round 2
Reviewer 2 Report (New Reviewer)
I thank the authors for the response and changes in the manuscript.
I only have a few more points.
1) Please check the grammar. For example the sentence
“Laparoscopic (LRP) and Robot-Assisted RP (RARP) are progressively emerged as the preferred options”. The phrase verb are progressively emerged should be are progressively emerging.
2) The Delta between POD 1 and POD 2 is significant? In the data look like there is no much difference. In that case delta POD3-2 could be changed to delta POD2-1 and reduce hospital time. Otherwise, it is ok as it is.
3) Regarding same day discharge. Is it possible perform it and measure red cell lines at POD 2 or 3? What is your recommendation or suggestion based on this study results?
Author Response
We thank the Reviewer for the pertinent and constructive comments, which significantly improved the manuscript.
- Regarding the sentence “Laparoscopic (LRP) and Robot-Assisted RP (RARP) are progressively emerged as the preferred options”: we changed the sentence according to the Reviewer’s suggestions in “Laparoscopic (LRP) and Robot-Assisted RP (RARP) are progressively emerging as the preferred options”. Moreover, we also corrected other minor grammar mistakes throughout the manuscript.
- Regarding the statistically significant differences between the value recorded at the POD1 and POD2: We agree with the Reviewer that the absolute value of Hb, RBC, and HCT recorded between the POD1 and the POD2 seems similar. However, we recorded a statistically significant difference (all p<0.001) between the two timing for all the red line cells parameter. That was the reason why we did not include this value in the table. Moreover, we decided to not include them to make the table easily readable to the reader.
- Regarding same-day discharge suggestion: Based on our results, we recommend discharging the patients as soon as possible to reduce health costs and increase patient compliance. However, the current study is not robust enough to strongly recommend the same discharge protocol. At now, even in case of same day discharge, we would recommend performing a blood test until POD 3 to be sure that the HB value would stabilize. However, in this way, the patient should or coming back to the hospital to take the blood sample or should perform it out of the hospital and send the result to us. This is why we prefer not to discharge the patients on the same day, but to keep them under our observation for three days. We would expect to change the patients’ management protocol in the future, after findings based on prospective results. We include this point in the discussion section, which now reads as follow:
“To date, in our daily clinical practice, we prefer to keep patients under our observations to facilitate the follow-up of blood test to ensure we dismiss patients in the safest conditions. We plan to change the patient management protocol in the future, after findings based on prospective and multicenter results.”
We hope that the Reviewer will find this reply satisfactory.
This manuscript is a resubmission of an earlier submission. The following is a list of the peer review reports and author responses from that submission.
Round 1
Reviewer 1 Report
Di Bello et al. should be congratulated for the work. These preliminary data could help the centers with low-volume surgery to improve their management of PCa patients undergoing RARP.
The manuscript is well written and easily readable, figures are understandable and clear. Concerning the manuscript, Authors wrote "The procedures were performed by three independent surgeons with high-volume experience in minimally invasive surgery.”. How many procedures were performed by the three surgeons, respectively? Were they homogeneously divided? Moreover, why were the blood tests recorded until the POD3? Authors should clarify that. A minor revision is required.